# Genetic Suppressor Element 1 (GSE1) Promotes the Oncogenic and Recurrent Phenotypes of Castration-Resistant Prostate Cancer by Targeting Tumor-Associated Calcium Signal Transducer 2 (TACSTD2)

**DOI:** 10.3390/cancers13163959

**Published:** 2021-08-05

**Authors:** Oluwaseun Adebayo Bamodu, Yuan-Hung Wang, Chen-Hsun Ho, Su-Wei Hu, Chia-Da Lin, Kai-Yi Tzou, Wen-Ling Wu, Kuan-Chou Chen, Chia-Chang Wu

**Affiliations:** 1Department of Urology, Shuang Ho Hospital, Taipei Medical University, New Taipei City 235, Taiwan; 16625@s.tmu.edu.tw (O.A.B.); 10352@s.tmu.edu.tw (S.-W.H.); 20500@s.tmu.edu.tw (C.-D.L.); 11579@s.tmu.edu.tw (K.-Y.T.); 15334@s.tmu.edu.tw (W.-L.W.); kuanchou@s.tmu.edu.tw (K.-C.C.); 2Department of Medical Research, Shuang Ho Hospital, Taipei Medical University, New Taipei City 235, Taiwan; d508091002@tmu.edu.tw; 3Department of Hematology and Oncology, Shuang Ho Hospital, Taipei Medical University, New Taipei City 235, Taiwan; 4Graduate Institute of Clinical Medicine, College of Medicine, Taipei Medical University, Taipei City 110, Taiwan; 5Department of Surgery, Division of Urology, Shin Kong Wu Ho-Su Memorial Hospital, Taipei City 111, Taiwan; m015695@ms.skh.org.tw; 6School of Medicine, College of Medicine, Fu-Jen Catholic University, New Taipei City 242, Taiwan; 7TMU Research Center of Urology and Kidney, Taipei Medical University, Taipei City 110, Taiwan; 8Department of Urology, School of Medicine, College of Medicine, Taipei Medical University, Taipei City 110, Taiwan

**Keywords:** prostate cancer, GSE1, TACSTD2, advanced disease, CRPC, castration resistance, therapy resistance, abiraterone, enzalutamide

## Abstract

**Simple Summary:**

In urological oncology clinics, worldwide, castration resistance and metastasis constitute a clinical quagmire and continue to hinder treatment success, despite the diagnostic and therapeutic advances of the last three decades. In this study, we present data that provide some preclinical evidence of the oncogenic role of dysregulated GSE1-TACSTD2 signaling, and show that the molecular or pharmacological targeting of GSE1 is a workable treatment strategy for inhibiting androgen-driven oncogenic signals, re-sensitizing cancerous cells to treatment, and repressing the metastatic-recurrent phenotypes of patients with prostate cancer.

**Abstract:**

Background: prostate cancer (PCa) is a principal cause of cancer-related morbidity and mortality. Castration resistance and metastasis are clinical challenges and continue to impede therapeutic success, despite diagnostic and therapeutic advances. There are reports of the oncogenic activity of genetic suppressor element (GSE)1 in breast and gastric cancers; however, its role in therapy resistance, metastasis, and susceptibility to disease recurrence in PCa patients remains unclear. Objective: this study investigated the role of aberrantly expressed GSE1 in the metastasis, therapy resistance, relapse, and poor prognosis of advanced PCa. Methods: we used a large cohort of multi-omics data and in vitro, ex vivo, and in vivo assays to investigate the potential effect of altered GSE1 expression on advanced/castration-resistant PCa (CRPC) treatment responses, disease progression, and prognosis. Results: using a multi-cohort approach, we showed that GSE1 is upregulated in PCa, while tumor-associated calcium signal transducer 2 (TACSTD2) is downregulated. Moreover, the direct, but inverse, correlation interaction between GSE1 and TACSTD2 drives metastatic disease, castration resistance, and disease progression and modulates the clinical and immune statuses of patients with PCa. Patients with GSE1^high^TACSTD2^low^ expression are more prone to recurrence and disease-specific death than their GSE1^low^TACSTD2^high^ counterparts. Interestingly, we found that the GSE1–TACSTD2 expression profile is associated with the therapy responses and clinical outcomes in patients with PCa, especially those with metastatic/recurrent disease. Furthermore, we demonstrate that the shRNA-mediated targeting of GSE1 (shGSE1) significantly inhibits cell proliferation and attenuates cell migration and tumorsphere formation in metastatic PC3 and DU145 cell lines, with an associated suppression of VIM, SNAI2, and BCL2 and the concomitant upregulation of TACSTD2 and BAX. Moreover, shGSE1 enhances sensitivity to the antiandrogens abiraterone and enzalutamide in vitro and in vivo. Conclusion: these data provide preclinical evidence of the oncogenic role of dysregulated GSE1–TACSTD2 signaling and show that the molecular or pharmacological targeting of GSE1 is a workable therapeutic strategy for inhibiting androgen-driven oncogenic signals, re-sensitizing CRPC to treatment, and repressing the metastatic/recurrent phenotypes of patients with PCa.

## 1. Introduction

Prostate cancer (PCa; International Classification of Diseases, ICD: C61) remains a principal cause of cancer-related morbidity and mortality. Globally, 1,414,259 newly diagnosed cases and 375,304 PCa-specific deaths were reported in the year 2020 alone, with a projected 1.72-fold increase in incidence and 1.97-fold increase in mortality worldwide by the year 2040 [1]. This may be associated with enhanced androgenic signaling, ensuing castration resistance and metastasis, a triad that continues to pose a significant challenge in urology clinics, especially impeding therapeutic success in spite of diagnostic and therapeutic advances [2,3,4]. Current data indicate that at least one in every three PCa cases will acquire a metastatic and/or recurrent phenotype within 48 months of the initial diagnosis [2,3]. One-fifth of these patients, with metastatic or recurrent disease, subsequently progress to castration-resistant PCa (CRPC) by the fifth year of clinical follow-up [2,3]. Unfortunately, after the development of castration resistance, the approximate median survival is a dismal 14 months [2,3].

Cumulative evidence indicates that androgen signaling plays a critical role in the pathogenesis and progression of prostate cancer [4]. Typically, bio-cellular events that facilitate the viability, survival, and proliferation of cancerous prostate cells are regulated by the androgen receptor (AR) directly or through the modulation of molecular mediators and downstream effectors [5]. Thus, the mainstay of treatment for metastatic PCa remains androgen-deprivation therapy (ADT), which inhibits AR activity, suppresses AR target genes, and elicits clinical remission that will last a couple of years [4,5]. However, because of the non-curative nature of ADT, progression to CRPC often follows, largely secondary to reactivated/restored androgen signaling [4,5]. To address this biological phenomenon, second-generation AR inhibitors (ARIs), such as abiraterone and enzalutamide, were developed to further suppress residual AR signaling and are the treatments of choice for patients with CRPC [6,7]. However, the clinical conundrum lingers, with a lack of durable complete remission (CR) and eventual treatment failure, despite an initial response and the promise of extended survival [6,7].

The early identification of patients at high risk of metastasis and/or recurrence after initial treatment may benefit clinical decision making and aid in the development of an effective treatment strategy that can improve prognosis. Over the past two decades, there has been an increase in biomarker exploration in the field of genitourinary oncology, with piqued interest in the molecular mechanisms underlying the roles of these biomarkers in oncogenicity, metabolic reprogramming, disease progression, and responses to therapy in patients with PCa [4,5]. This molecular renaissance, hinged on the discovery and validation of novel diagnostic or prognostic biomarkers, continues to facilitate the elucidation of the pathogenesis and biology of PCa and concomitantly facilitates patient stratification into responders or non-responders to certain therapies, as well as discriminating ‘progressors’ from ‘non-progressors’ [4,5,6,7]. Against the background of the unabated incidence, high mortality burden, and increased odds of disease progression, as well as a largely unclear underlying bio-mechanism of progression, the present study probed actionable biomarkers that could provide clinically objective measures of PCa biology, improve patient stratification, and inform therapeutic decision making.

Recently, it was reported that the relatively unknown genetic suppressor element 1 (GSE1, KIAA0182), a proline-rich protein with coiled-coil domains, was overexpressed in patients with breast cancer and associated with poor prognosis; targeting GSE1 elicited the upregulation of miR-489-5p, with the repression of breast cancer cell proliferation, migration, and invasion [8]. In another recent study, it was shown that the expression of GSE1 was upregulated in patients with gastric cancer and, concomitant with enhanced SLC7A5 expression, was implicated in increased tumor growth, metastasis, trastuzumab resistance, and worsened postoperative survival outcomes [9,10]. Our evolving understanding of the oncogenic activity of GSE1, coupled with its largely unexplored role in the metastasis, therapy resistance, and susceptibility to disease recurrence in PCa patients, informed the present study.

The last decade has been characterized by the increased evaluation of the biomolecular role of the 40 kDa tumor-associated calcium signal transducer 2 (TACSTD2, also known as trophoblast cell surface antigen 2, TROP2) in tumor initiation and progression. The intron-deficient, epithelial cell adhesion molecule TACSTD2 is a ubiquitously expressed glycoprotein, and its expression is associated with stem cell-defining attributes, including ‘regenerative capacity in various tissues’ [11,12,13].

The epithelium of the adult prostate contains three distinct cell types: basal, luminal, and neuroendocrine [14]. In this context, there is accruing evidence of the tissue regenerative activity of CD49f + Sca1 + basal cells from the tripartite adult prostate epithelium; interestingly, TACSTD2 is enriched in these basal cells and exhibits stem cell-/progenitor cell-like traits, such as ‘localization to the region of the gland proximal to the urethra and enrichment for sphere-forming and colony-forming cells’ [14]. However, while the overexpression of TACSTD2 has been described in several cancer types, conflicting reports abound, with functional studies showing not only oncogenic but also tumor suppressor roles [11]. Interestingly, despite this implication of TACSTD2 in pluripotency and contemporary knowledge that TACSTD2^high^ basal cells efficiently form spheres in vitro, the role of TACSTD2 in prostate cancerization and therapy response is under-explored or, rather, unclear.

Herein, we present preclinical evidence of the role of GSE1/TACSTD2 in signaling as an indicator of disease course and as a putative biomarker of the therapy responses in patients with PCa. These findings also suggest the clinical feasibility of targeting GSE1 as an efficacious therapeutic strategy to re-sensitize metastatic/recurrent CRPC cells to ADT or as antiandrogen therapy (abiraterone or enzalutamide) while mitigating susceptibility to disease recurrence.

## 2. Material and Methods

### 2.1. Prostate Cancer Tissue Samples

Prostate cancer tissue samples (*n* = 56) were obtained from the Taipei Medical University Shuang Ho Hospital tissue bank, following ethical approval from the Institutional Review Board of the Taipei Medical University (approval number: N202101071). The requirement for patients’ signed informed consent was waived because of the retrospective nature of the study.

### 2.2. Cell Culture

The normal human primary prostate epithelial HPrEC (ATCC^®^ PCS-440-010™) cell line, LNCaP (ATCC^®^ CRL-1740™; 5-a dihydrotestosterone-responsive, androgen-dependent, and metastatic), PC-3 (ATCC^®^ CRL-1435™; low acid phosphatase and testosterone-5-a reductase, androgen-independent, and metastatic), and the hormone-insensitive DU145 (ATCC^®^ HTB-81™) prostate carcinoma cell line were obtained from the ATCC (American Type Culture Collection, Manassas, VA, USA). The cells were cultured in RPMI 1640 (Thermo Fisher Scientific Inc., Bartlesville, OK, USA), supplemented with 10% fetal bovine serum (FBS, #26140079, Thermo Fisher Scientific Inc., Bartlesville, OK, USA) and 100 U/mL penicillin–streptomycin (Thermo Fisher Scientific Inc., Bartlesville, OK, USA). The cells were subcultured at ≥98% confluence, and the growth media were changed every 72 h.

### 2.3. Antibodies and Reagents

Monoclonal antibodies against GSE1 (#sc-514946), TACSTD2 (#sc-376746), vimentin (#sc-66002), SLUG/SNAI2 (#sc-166476), BAX (#sc-7480), BCL2 (#sc-7382), VEGF (#sc-7269), OCT3/4 (#sc-5279), MDR1/ABCB1 (#sc-13131), and GAPDH (#sc-32233) were purchased from Santa Cruz Biotechnology Inc. (Santa Cruz, CA, USA). Abiraterone (CB-7598, #S1123, ≥99% (HPLC)) and enzalutamide (MDV3100, #S1250, ≥99% (HPLC)) were purchased from Selleck Chemicals (Houston, TX, USA). Stock solutions of 100 mM in 0.01% dimethyl sulfoxide (DMSO, #276855, Sigma-Aldrich^®^, Merck KGaA, Darmstadt, Germany) were stored at −20 °C until use.

### 2.4. Knockdown of GSE1 by shRNA Interference

A GSE1 shRNA plasmid (h) (#sc-93036-SH, Santa Cruz Biotechnology Inc., Santa Cruz, CA, USA) was used to knock down GSE1 in cells, strictly following the manufacturer’s protocol. Briefly, after growing PC-3 or DU145 cells to 70% confluence in 6-well plates, 1 µg/10 µL of re-suspended GSE1shRNA plasmid DNA was diluted in 90 µL of antibiotic-free shRNA Plasmid Transfection Medium (#sc-108062, Santa Cruz Biotechnology Inc., Santa Cruz, CA, USA) for each transfection. After 72 h of incubation, stably transfected cells were selected with 2 μg/mL puromycin (#sc-108071, Santa Cruz Biotechnology Inc., Santa Cruz, CA, USA), as recommended by the manufacturer. GSE1 knockdown in the cells was verified by Western blot analysis.

### 2.5. Cell Viability and Proliferation Colorimetric Assay

The sulforhodamine B (SRB) assay was performed to assess cell viability. Wild-type (WT) or GSE1-silenced (shGSE1) PC-3, and DU145 cells were seeded at 3 × 10^3^ cells per well in 96-well plates containing complete growth medium with or without the indicated concentration of abiraterone or enzalutamide and incubated in humidified 5% CO_2_ at 37 °C for 48 h. Thereafter, the cell viability was evaluated following the manufacturer’s instructions. Briefly, WT or shGSE1 cells were fixed with 10% trichloroacetic acid (TCA: #T6399, Sigma-Aldrich^®^, Merck KGaA, Darmstadt, Germany), carefully washed with ddH_2_O, and then stained with 0.4:1 (*w/v*) SRB–acetic acid solution (#230162, Sigma-Aldrich^®^, Merck KGaA, Darmstadt, Germany). Unbound SRB dye was carefully washed off the cells with 1% acetic acid, the plates were air-dried, and the bound SRB dye was solubilized in 10 mM Tris base (#3163, Tocris Bioscience, Avon, UK). To analyze cell proliferation, we used the Invitrogen alamarBlue™ High-Sensitivity Cell Viability Reagent (#A50100, Thermo Fisher Scientific Inc., Bartlesville, OK, USA), strictly following the manufacturer’s instructions. Briefly, after seeding PC-3 and DU145 WT (or shGSE1) cells in triplicate, with each assay having three biological replicates, at the indicated time point (Day 2), the cells were incubated with alamarBlue™ at 37 °C for 2 h. The number of dye-stained proliferating cells was measured at an absorbance wavelength of 570 nm in a Molecular Devices SpectraMax M3 Multi-Mode Microplate Reader (Molecular Devices LLC., San Jose, CA, USA).

### 2.6. Cancer Data Set Retrieval

The public and free-access online cancer data repositories used in this study include The Cancer Genome Atlas (TCGA), Gene Expression Omnibus (GEO), and Cancer Cell Line Encyclopedia (CCLE). The TCGA dataset used was the prostate adenocarcinoma (PRAD) IlluminaHiSeq RNAseq data (*n* = 623), which were downloaded and analyzed using the National Cancer Institute Genomic Data Commons Data Portal (https://portal.gdc.cancer.gov/, accessed on 9 December 2020) and the Oncomine interface (https://www.oncomine.org/resource/main.html#v:18, accessed on 5 December 2020). The GSE35988 (*n* = 122), GSE6099 (*n* = 101), GSE32265 (*n* = 55), GSE16560 (*n* = 281), GSE21887 (*n* = 12), GSE21032 (*n* = 281), GSE109708 (*n* = 8), and GSE104935 (*n* = 10) datasets were all retrieved from the GEO online platform (https://www.ncbi.nlm.nih.gov/geo/, accessed on 9 December 2020).

### 2.7. Immunohistochemical (IHC) Staining Assay

Immunohistochemical (IHC) analysis was performed on formalin-fixed, paraffin-embedded (FFPE) sections from our PCa cohort, which consisted of samples from patients with different tumor grades (*n* = 56; normal-like: Gleason score (GS) ≤ 5; low: GS = 6; medium: GS = 7; high: GS ≥ 8), following ethical approval by the Taipei Medical University Institutional Review Board (approval number: N202101071) and compliant with recommendations from the Declaration of Helsinki for biomedical research involving human subjects. Samples were probed with antibodies against GSE1, TACSTD2, OCT3/4, and ABCB1/MDR1 at 1:200 dilutions, following the standard IHC protocol. The immunoreactivity, based on the total stained area, stained cell count, average size of the stained area, percentage stained area, and perimeter was quantified using the National Institutes of Health ImageJ software version 1.49 (https://imagej.nih.gov/ij/).

### 2.8. Western Blotting Assay

Twenty micrograms of WT, or shGSE1 PC-3, and DU145 cell protein samples were separated using 10% sodium dodecyl sulfate (SDS)–polyacrylamide gel electrophoresis (PAGE). The separated proteins were transferred onto polyvinylidene fluoride (PVDF) membranes using the Bio-Rad Mini-Protein electro-transfer system (Bio-Rad Laboratories, Inc., Hercules, CA, USA). The PVDF membranes were blocked with 5% non-fat milk in Tris-buffered saline with Tween 20 (TBST) for 1 h and then incubated with primary monoclonal antibodies against GSE1 (1:1000, Santa Cruz Biotechnology), TACSTD2 (1:1000, Santa Cruz Biotechnology), vimentin (1:1000, Santa Cruz Biotechnology), SLUG/SNAI2 (1:1000, Santa Cruz Biotechnology), BAX (1:1000, Santa Cruz Biotechnology), BCL2 (1:1000, Santa Cruz Biotechnology), VEGF (1:1000, Santa Cruz Biotechnology), OCT3/4 (1:1000, Santa Cruz Biotechnology), MDR1/ABCB1 (1:1000, Santa Cruz Biotechnology), and GAPDH (1:1000, Santa Cruz Biotechnology) overnight at 4 °C. Thereafter, the PVDF membranes were incubated with horseradish peroxidase (HRP)-conjugated secondary antibodies for 1 h at room temperature and washed thrice with cold 1X phosphate-buffered saline (PBS, #11666789001, Sigma-Aldrich^®^, Merck KGaA, Darmstadt, Germany); then, the protein bands were detected using an enhanced chemiluminescence detection system (Thermo Fisher Scientific Inc., Waltham, MA, USA). Densitometry was performed with the National Institutes of Health ImageJ software version 1.49 (https://imagej.nih.gov/ij/).

### 2.9. Scratch Wound-Healing Assay

A scratch wound-healing assay was performed to assess cell migration. Briefly, WT, or shGSE1 PC-3, and DU145 cells were seeded and cultivated in 6-well plates (Corning, Corning, NY, USA) containing complete growth medium with 10% FBS. The medium was changed to low-serum (1% FBS) growth medium when the cells reached >98% confluence. The median axes of the single-layered, adherent cells were scratched using sterile yellow pipette tips. The cell migration, based on the closure of the scratch wounds, was monitored over time, and images were captured at 0 and 24 h post-denudation under a light microscope using a 10X objective lens. Thereafter, the images were analyzed using the National Institutes of Health ImageJ software version 1.49 (https://imagej.nih.gov/ij/).

### 2.10. Tumorsphere Formation Assay

WT, or shGSE1 PC-3, and DU145 cells were seeded at 5 × 10^4^ cells per well in ultra-low-attachment 6-well plates (Corning, Corning, NY, USA) containing RPMI 1640, supplemented with 20 ng/mL basic fibroblast growth factor (bFGF; #13256029, Invitrogen), GibcoTM B-27TM supplement (#17504044, Invitrogen, Carlsbad, CA, USA), and 20 ng/mL epidermal growth factor (EGF; #PHG0311, Invitrogen). The PCa cell lines were cultured at 37 °C in a humidified 5% CO_2_ incubator for 5–7 days. The formed tumorspheres with sizes ≥ 100 µm were counted under an inverted phase-contrast microscope.

### 2.11. Tumor Xenograft In Vivo Studies

For in vivo tumor xenograft studies, 1 × 10^6^ PC-3_WT, or PC-3_shGSE1, cells in 100 μL of complete growth medium were subcutaneously injected into the right flanks of 7–8-week-old male BALB/c-nu mice (28.3 ± 5.2 g; *n* = 5 per group) (BioLASCO, Taipei City, Taiwan). The mice were randomly divided into the control (PC-3_WT) and test (PC-3_shGSE1, enzalutamide, and PC-3_shGSE1+enzalutamide) groups. For the treatment group, 10 mg/kg/day of enzalutamide, administered intraperitoneally (ip) every 72 h for 4 weeks, was initiated as soon as the tumors became palpable (tumor volume ~100 mm^3^). For the control (positive: PC-3_WT, or negative: PC-3_shGSE1) group, 100 μL/day of vehicle 0.01% DMSO was administered ip every 72 h for 4 weeks. Tumor growth was monitored throughout the experiment by taking caliper measurements of tumors twice weekly, and the tumor volume was estimated using the following formula: ½ [length (mm)] × [width (mm)]^2^. The mice were humanely sacrificed at the end of the study on day 30. The tumors were excised and carefully analyzed, and tumor samples were used for subsequent assays. The animal studies complied with the approved protocol of the Lab Animal Committee/Institutional Animal Care and Use Committee (Approval no.: LAC-2020-0553) of Taipei Medical University.

### 2.12. Statistical Analysis

All the data are expressed as the mean ± standard deviation (SD) for assays performed at least 3 times independently. Two-sided Student’s *t*-tests were used for comparison between 2 groups, while one-way ANOVA with Tukey’s post-hoc tests were used for comparing ≥3 groups. Kaplan–Meier survival analyses aided in the comparison of survival rates between the control and test groups. The Pearson chi-square (*X*^2^) test was used for correlation analysis and the determination of association. All the statistical analyses were performed using GraphPad Prism version 8.0.0 for Windows (GraphPad Software, La Jolla, CA, USA). The *p*-values < 0.05 were considered statistically significant.

## 3. Results

### 3.1. Increased GSE1/TACSTD2 Expression Ratio Defines Patients with Prostate Cancer

Our bioinformatics-aided probe of the Gene Expression Omnibus (GEO)-derived GSE35988 dataset of lethal castration-resistant PCa (CRPC) (Grasso Prostate, *n* = 122) showed that, compared to normal prostate gland samples, samples from patients with PCa exhibited a higher expression of *GSE1* mRNA (fold change = 1.12, *p* = 0.03) (Figure 1A). Similarly, from a reanalysis of the GSE6099 microarray gene expression profiling of the PCa progression dataset (Tomlins Prostate, *n* = 101), we observed that PCa cases had a 2.36-fold increase (*p* = 8.22 × 10^−6^) in *GSE1* transcript expression, relative to that in normal prostate glands (Figure 1B). On the other hand, we found that *TACSTD2* expression was downregulated 1.23-fold (*p* = 0.14) and 1.47-fold (*p* = 0.87) in the PCa group, compared to that in their normal prostate gland counterparts from the Grasso Prostate and Tomlins Prostate cohorts, respectively (Figure 1C,D). This was further corroborated by results from the analysis of the National Cancer Institute Genomic Data Commons (NCI GDC) TCGA PRAD cohort (*n* = 623), which showed that, while the *GSE1* gene was significantly upregulated (1.36-fold, *p* = 1.88 × 10^−7^), the *TACSTD2* gene expression was downregulated (0.98-fold, *p* = 2.58 × 10^−1^) in PCa, compared with the normal samples (Figure 1E,F). These data indicate, at least in part, that an increased GSE1/TACSTD2 expression ratio defines patients with PCa.

### 3.2. The Interaction between GSE1 and TACSTD2 Drives Metastatic Disease, Castration Resistance, and Disease Progression in Patients with Prostate Cancer

Having demonstrated an inverse correlation between GSE1 and TACSTD2, we further probed its biomolecular and clinical implications. Using the TCGA PRAD cohort (*n* = 623), we found that, compared with the non-tumor ‘normal’ and primary PCa cases, patients with metastatic PCa exhibited a significantly higher expression of *GSE1* (normal < tumor < metastatic, *p* = 1.11 × 10^−6^) (metastatic vs. normal: 2.01-fold, *p* = 3.91 × 10^−1^; metastatic vs. tumor: 1.36-fold, *p* = 4.64 × 10^−2^) (Figure 2A). Conversely, *TACSTD2* gene expression was the lowest in the metastatic samples, compared with the normal or tumor samples (metastatic vs. normal: 1.39-fold, *p* = 5.77 × 10^−2^; metastatic vs. tumor: 0.72-fold, *p* = 1.09 × 10^−1^) (Figure 2B). Because of the interplay between disease aggression, progression, and prognosis, we reanalyzed the TCGA PRAD cohort (*n* = 623) for a potential correlation between the differential expression of GSE1 and TACSTD2 and the metastasis (M) stage. We observed that, compared with its expression in patients with negative metastasis status (M0), high *GSE1* expression was mostly associated with metastasis to the bones (M1b) in patients with PCa (f = 0.70, *p* = 0.55) but less so with metastasis to distant lymph nodes (M1a) and distant organs (M1c) (Figure 2C). Interestingly, we also found that, relative to the M0 cases, the expression of the *TACSTD2* gene was significantly downregulated in patients with metastasis to the bones (M1b) and to distant lymph nodes (M1a) (f = 1.23, *p* = 0.30) (Figure 2D). Moreover, we showed that the levels of *GSE1* and *TACSTD2* expression were inversely correlated (Pearson’s rho = −0.18, *p* = 0.00004) and that *GSE1*^high^*TACSTD2*^low^ was associated with a higher incidence of biochemical recurrence (BCR) (Figure 2E). The principal component analysis (PCA) of the GSE32265 expression data, for primary localized PCa vs. castration-resistant bone metastatic prostate (*Homo sapiens*, A-AFFY-33, AFFY_HG_U133A; *n* = 55 samples; 22 283 genes), showed that GSE1^low^TACSTD2^high^ mostly explained ‘normal’ non-PCa cases, whereas GSE1^high^TACSTD2^low^ largely characterized castration-resistant bone metastatic prostate cancer, especially after androgen deprivation and ADT-naive localized PCa (Figure 2F). Our expression-based heatmap, generated from the analysis of the same GSE32265 cohort (*n* = 55), showed that the high expression of GSE1 correlated with the upregulated expression of the angiogenesis biomarkers (vascular endothelial growth factor A (VEGFA) and angiopoietin 2 (ANGPT2)), a biomarker of T-cell non-inflamed or cold tumors (β-catenin (CTNNB1)), metastasis biomarkers (slug (SNAI2) and vimentin (VIM)), and a marker of proliferation (MKI67/Ki-67), with concomitantly suppressed TACSTD2 expression in castration-resistant bone metastatic prostate cancer, regardless of the presence of the transmembrane serine protease (TMPRSS)2-erythroblast transformation-specific transcription factor ERG variant 10 (ERG) fusion gene (Figure 2G). Interestingly, we also found the co-upregulation of GSE1, ANGPT2, VEGFA, KLK3 (kallikrein-3, prostate-specific antigen), CTNNB1, and TACSTD2 in localized PCa in the presence of the TMPRSS2-ERG fusion gene (Figure 2G). In parallel Western blot assays, we also demonstrated that, compared to HPrEC (normal human prostate epithelial cells), GSE1 protein expression was concomitantly upregulated with VIM, SNAI2, and VEGFA in the LNCaP (androgen-sensitive metastatic PSA^positive^), PC3 (bone metastatic grade IV androgen-independent), and DU145 (moderately metastatic androgen-independent PSA^negative^) cells, while TACSTD2 expression was downregulated (Figure 2H). These data show that the interaction between GSE1 and TACSTD2 plays a critical role in metastatic disease, castration resistance, and disease progression in patients with PCa.

### 3.3. Inversely Correlated GSE1 and TACSTD2 Expression Patterns Predicts Survival of Patients with Prostate Cancer

Having shown that the interaction between GSE1 and TACSTD2 plays a critical role in metastatic disease, castration resistance, and disease progression in patients with PCa, we probed the GSE16560/GPL5474 PCa disease progression dataset (*n* = 281); consistent with previous data, we observed an interesting expression pattern wherein the GSE1 expression ‘peak’ corresponded to a TACSTD2 expression ‘dip’ (Figure 3A). Similarly, our three-dimensional visualization of the gene expression data points (individual samples) in the same GSE16560/GPL5474 PCa cohort (*n* = 281) showed that the TACSTD2 ‘knuckle/bulge’ finely fits into the GSE1 ‘hollow/gorge’, further corroborating an inverse GSE1/TACSTD2 correlation (Figure 3B). To further investigate the clinical relevance of this GSE1/TACSTD2 expression profile, we reanalyzed a pooled PCa dataset (*n* = 2205 samples) consisting of prostate cancer (DKFZ, Cancer Cell (2018)), prostate adenocarcinoma (MSKCC/DFCI, Nature Genetics (2018)), metastatic castration-sensitive prostate cancer (MSK, Clin Cancer Res (2020)), and metastatic prostate adenocarcinoma (SU2C/PCF Dream Team, PNAS (2019)) from the cBioPortal platform (https://www.cbioportal.org/, accessed on 13 December 2020). The results showed that patients with high *GSE1* expression exhibited markedly worse overall survival (OS) than their counterparts with high *TACSTD2* expression (Figure 3C). We also observed that, while *TACSTD2* mRNA expression was positively correlated with the overall survival time (Spearman rho = 0.12, *p* = 0.31) (Figure 3D), *GSE1* expression was inversely correlated (Spearman rho = −0.07, *p* = 0.55) in the pooled PCa data (Figure 3E). In parallel analyses, using the TCGA PRAD cohort (*n* = 623), we showed that patients with high *GSE1* exhibited worse OS (hazard ratio, HR = 3.3; Logrank *p* = 0.14) and worse disease-free survival (DFS: HR = 0.87; Logrank *p* = 0.66), compared to the low *GSE1* group (Figure 3F). Conversely, high *TACSTD2* expression conferred better OS (HR = 0.62, *p* = 0.57) and DFS (HR = 0.56, *p* = 0.04) (Figure 3G). Consistent with this, the survival map, generated from the analysis of the TCGA PRAD data, showed a strong association between the GSE1^high^TACSTD2^low^ profile and disease-specific death, while the DFS was strongly associated with the GSE1^low^TACSTD2^high^ profile (Figure 3H). These data indicate that the inversely correlated GSE1 and TACSTD2 expression patterns determine the survival of patients with prostate cancer.

### 3.4. The GSE1 and TACSTD2 Signal Interplay Affects the Clinical and Immune Statuses of Patients with Prostate Cancer

Premised on our understanding that “copy-number alterations robustly define clusters of low- and high-risk disease beyond that achieved by Gleason score” [15], to gain better insight into the role of GSE1/TACSTD2 signaling in the clinical course of PCa, we employed the integrative genomic clustering of the Memorial Sloan Kettering Cancer Center (MSKCC; GSE21032) dataset (*n* = 281), as described in [15], and found that *GSE1* expression was higher in iClusters 1, 3, 4, and 5 (1 > 5 > 3 > 4), which are associated with a very unfavorable prognosis, than in iClusters 2 and 6, which are associated with a very favorable prognosis [15,16] (Figure 4A). Conversely, *TACSTD2* expression was lower in iClusters 1 and 5 than in iClusters 2, 3, 4, and 6 (Figure 4B). Moreover, we showed that *GSE1* expression was highest in patients with Gleason scores of 8 (4 + 4 > 3 + 5), intermediate in those with Gleason scores of 6 (3 + 3), and lowest in patients with Gleason scores of 7 and 9 (Figure 4C). Conversely, *TACSTD2* expression was highest in Gleason score 6 and 7 cases, intermediate in those with Gleason scores of 8, and lowest in those with Gleason scores of 9 (Figure 4D). Using the same MSKCC cohort (*n* = 281), we observed a time-dependent, ambivalent association between differentially-expressed GSE1 and biochemical recurrence-free survival (*p* = 0.64) (Figure 4E). However, high TACSTD2 was associated with a 39.7% reduction in biochemical, recurrence-free survival by Year 5 (*p* = 0.009) (Figure 4F). Furthermore, because of the critical role of the intratumoral immune cell infiltration level in disease progression and prognosis, we also investigated the potential correlation of GSE1 and TACSTD2 expression with the immune cell infiltration level in the TCGA PRAD cohort (*n* = 623). Our analyses revealed that, while *GSE1* gene expression was positively correlated with tumor purity (correlation, cor = 0.13; *p* = 0.0085), *TACSTD2* expression was inversely correlated (cor = −0.08, *p* = 0.11), suggesting a probable high expression of *TACSTD2* in the tumor microenvironment (TME), while *GSE1* was highly expressed in tumor cells (Figure 4G). Interestingly, consistent with current knowledge that intratumoral B-cell infiltration is higher in PCa than the extratumoral B-infiltrates in adjacent benign prostate tissue regions [17], we found that *GSE1* expression was 8.29-fold more correlated with B-cell infiltration than *TACSTD2* expression (*GSE1*: partial cor = 0.26, *p* = 1.24 × 10^−7^, vs. *TACSTD2*: partial cor = 0.03, *p* = 0.53) (Figure 4G). Similarly, relative to that between *TACSTD2* expression and tumor-infiltrating CD8+ T cells, we found a 1.26-fold higher correlation between *GSE1* expression and tumor-infiltrating CD8+ T cells (Figure 4G). This is concordant with the notion that tumor-infiltrating CD8+ T cells exhibit enhanced expression of programmed cell death protein 1 (PDCD1), an immune-inhibitory receptor associated with an “exhausted” CD8+ T-cell phenotype [18]. We also found that the *GSE1* and *TACSTD2* expression levels had partial correlations of 0.18 (*p* = 1.72 × 10^−4^) and 0.15 (*p* = 2.13 × 10^−3^), respectively, with tumor-infiltrating or associated macrophages, and this is consistent with contemporary knowledge that tumor-infiltrating, or associated macrophages, are highly enriched in aggressive cancer subtypes, orchestrate stromal oncogenic signaling, and predict worse prognosis [19,20] (Figure 4G). In line with reports that the size of the tumor-infiltrating neutrophil (TIN) pool is implicated in cancer aggression and is touted as a useful predictor of chemotherapy responses or prognosis in different cancer types [21], we observed that TIN exhibited partial correlations of 0.16 (*p* = 1.59 × 10^−3^) and 0.19 (*p* = 1.42 × 10^−4^) with *GSE1* and *TACSTD2* expression levels, respectively (Figure 4G). These data indicate that GSE1 and TACSTD2 signal interplay affects the clinical courses and immune statuses of patients with PCa.

### 3.5. GSE1 and TACSTD2 Expression Profile Is Associated with Therapy Responses and Clinical Outcomes in Patients with Prostate Cancer

Against the background of our earlier result, suggesting the involvement of GSE1 and TACSTD2 expression in the disease course, therapy response, and prognosis, we further probed this possibility for further clarification. Reanalysis of the TCGA PRAD cohort data (*n* = 623) showed that the *GSE1* gene was least expressed in patients with stable disease and showed the strongest expression in those with progressive disease (stable disease < complete response < partial response < progressive disease; f-stat = 2.86, *p* = 0.04) (Figure 5A). Conversely, the *TACSTD2* expression level was the lowest in stable and progressive disease, intermediate in that with a partial response, and the highest in patients with complete responses/remission (stable disease < progressive disease < partial response < complete response; f-stat = 0.46, *p* = 0.71) (Figure 5B). Moreover, the principal component analysis of the GSE32265 dataset, regarding the expression profiles of primary localized PCa and castration-resistant bone metastatic prostate (*Homo sapiens*, A-AFFY-33, and AFFY_HG_U133A; *n* = 55 samples; 22 283 genes), showed that GSE1^low^TACSTD2^high^ mostly explained ‘normal’, non-PCa cases and some localized PCa, whereas GSE1^high^TACSTD2^low^ largely characterized castration-resistant bone metastatic prostate cancer, especially after androgen deprivation and ADT-naive localized PCa (Figure 5C). Of therapeutic relevance, a heatmap (based on expression profiles in the GSE32265 cohort (*n* = 55)) showed that, while TACSTD2 expression was downregulated, GSE1 expression was co-upregulated with biomarkers of drug metabolism/resistance, multidrug resistance-associated protein 1 (MRP1/ABCC1), cytochrome P450 family 3 subfamily A member 5 (CYP3A5), CYP3A4, a biomarker of T-cell non-inflamed or cold tumors (β-catenin (CTNNB1)), and stemness markers (BMI1, POU5F1/OCT4, and SOX2) in castration-resistant bone metastatic PCa exposed to ADT, regardless of the TMPRSS2-ERG fusion gene status (Figure 5D). In ADT-naive localized PCa, we also found similar concomitant upregulation of GSE1 with CYP3A5, SOX2, KLF4, BMI1, ABCC1, and MDR1/ABCB1 (Figure 5D). Furthermore, principal component analysis of the GSE21887 dataset, regarding potential targets for the treatment of CRPC (*Homo sapiens*, A-AFFY-44 and AFFY_HG_U133A_PLUS_2; *n* = 12 samples; 54 675 genes), showed that GSE1^high^TACSTD2^low^ mostly characterized castration-induced regression nadirs (or ADT failure), while GSE1^low^TACSTD2^high^ largely explained androgen-dependent/sensitive tumor growth; however, it is therapeutically relevant that an intriguing third group, GSE1^high^TACSTD2^high^, characterized patients with castration-resistant tumor regrowth (and, by inference, disease recurrence) (Figure 5E). Our heatmap revealed that androgen-dependent growth was associated with the concurrent upregulation of GSE1, ABCC1/MRP1, ABCB1/MDR1, SOX2, POU5F1/OCT4, and CYP3A5. For the castration-induced regression nadir group (i.e., those with the weakest or most unsuccessful castration-induced tumor regression), GSE1, CYP3A5, CTNNB1, CYP3A4, POU5F1, ABCB1, alkaline phosphatase liver/bone/kidney isozyme (ALPL), PROM1/CD133, ABCG2, and SOX2 were concomitantly upregulated. We also found the co-upregulation of GSE1, SOX2, BMI1, KLF4, CTNNB1, and TACSTD2 in castration-resistant regrowth/recurrent cases (Figure 5F). These data indicate that GSE1 and TACSTD2 expression profiles are indicative of therapy responses and clinical outcomes in patients with PCa.

### 3.6. GSE1 and TACSTD2 Interaction or Expression Profiles Reflect Abiraterone/Enzalutamide Drug Resistance, Androgen Sensitivity, and Castration Resistance in Patients with Prostate Cancer

Having shown that the GSE1 and TACSTD2 expression profiles influence the therapy responses and clinical outcomes in patients with PCa, we performed immunohistochemical staining of the samples from our in-house PCa cohort (*n* = 56). Consistent with the data above, we found that, compared with that in the non-cancerous prostate gland (‘normal’) samples, the expression of the GSE1 protein increased with the tumor grade, with the strongest immunoreactivity in patients with medium- and high-grade PCa; conversely, TACSTD2 expression was the highest in the ‘normal’ prostate samples and barely expressed in patients with high-grade PCa (Figure 6A). Next, to understand the molecular linkage between GSE1 and TACSTD2, we performed a molecular connectivity analysis using the Schrödinger PyMOL 2.5 molecular visualization system (https://pymol.org/2/, accessed on 9 January 2021). Our molecular docking showed that GSE1 directly binds to TACSTD2 to form a GSE1/TACSTD2 complex (docking score = −251.54, complementarity score = 15,730, complex interface area = 2792.70, atomic contact energy ACE = −509.95 kcal/mol, and clustering root-mean-square deviation, RMSD = 4 Å) (Figure 6B). The 3D transformation data, consisting of three rotational angles (2.08°, −0.55°, and 3.07°) and three translational parameters (456.76°, −418.70°, and 329.61°), were applied to the ligand molecule, GSE1 (Figure 6B; also, see Appendix A). Moreover, to determine the pharmacologic dependency of current antiandrogen therapy on GSE1/TACSTD2 signaling, using the Cancer Dependency Map platform (https://depmap.org/portal/, accessed on 28 December 2020), we demonstrated that enzalutamide sensitivity in treated 22RV1 (primary), DU145 (metastatic), PC3 (metastatic), and LNCaP clone FGC (metastatic) cell lines was inversely correlated with GSE1 expression (Pearson rho = −0.78, *p* = 0.43), but positively correlated with TACSTD2 expression (Pearson rho = 0.59, *p* = 0.60) (Figure 6C). Reanalysis of the GSE104935 dataset for enzalutamide (*n* = 10) showed that, compared with that in the enzalutamide-resistant samples, *TACSTD2* expression was higher in enzalutamide-sensitive samples (1.8-fold, *p* = 0.30). Conversely, relative to that in the enzalutamide-resistant samples, *GSE1* expression was low in the enzalutamide-sensitive samples (0.58-fold, *p* = 0.25) (Figure 6D). Using the GSE109708 dataset (*n* = 8), compared with the pre-castration androgen sensitivity, we observed a mild increase in the GSE1/TACSTD2 ratio in the abiraterone/enzalutamide-resistant samples (1.07-fold, *p* = 0.052); however, this increase was significantly enhanced in the CRPC cases (3.04-fold, *p* = 0.0009) (Figure 6E). Corroborating these results, statistical analysis of the GSE150895 dataset (*n* = 6) showed that, while the TACSTD2/GSE1 ratio was higher in the enzalutamide-sensitive cases, the GSE1/TACSTD2 ratio was higher in the enzalutamide-resistant samples (Figure 6F). These data indicate that the GSE1 and TACSTD2 interaction and expression profiles reflect abiraterone/enzalutamide drug resistance, androgen sensitivity, and castration resistance in patients with PCa.

### 3.7. Targeting GSE1 Signaling Suppresses Metastatic and Cancer Stemness Phenotypes and Enhances Sensitivity to Abiraterone or Enzalutamide in Metastatic Castration-Resistant Prostate Cancer In Vitro and In Vivo

Having shown that the GSE1 and TACSTD2 interaction and expression profiles reflect abiraterone/enzalutamide drug resistance, androgen sensitivity, and castration resistance in patients with PCa, we assessed the potential effect of targeting GSE1/TACSTD2 signaling through the shRNA-mediated inhibition of GSE1 (shGSE1). Our results show that shGSE1 significantly downregulated the expression of the GSE1 (knockdown efficacy: PC3_shGSE1 = 82.6%; DU145_shGSE1 = 89.4%), VIM, SNAI2, and BCL2 proteins, while concomitantly upregulating the TACSTD2 and BAX protein expression levels in the PC3 and DU145 cell lines (Figure 7A). Compared to the wild-type (WT) cells, silencing GSE1 also suppressed the proliferation of the PC3_shGSE1 and DU145_shGSE1 cells by 2.78-fold (*p* < 0.001) and 2.54-fold (*p* < 0.001) (Figure 7B). Our migration assays also showed that, compared with that of the PC3_WT cells, at the 24 h time point, the migration of the PC3_shGSE1 cells was significantly attenuated (2.89-fold, *p* < 0.001) (Figure 7C). We also investigated the effect of targeting GSE1 in tumorspheres, which are in vitro models of cancer stem cells (CSCs), and demonstrated that the tumorsphere-formation capability of the PC3_shGSE1 and DU145_shGSE1 cells was markedly suppressed, in comparison to that of their WT counterparts (Figure 7D). Moreover, we examined if, and to what extent, shGSE1 affected antiandrogen treatment. We found that shGSE1 significantly enhanced the anticancer/killing effects of abiraterone and enzalutamide (Figure 7E). Furthermore, we investigated the replicability of these potentiated anticancer effects in vivo using BALB/c-nu mice. We observed that, compared to that in the vehicle-treated PC3_WT-inoculated control group, tumor growth in the PC3_WT-inoculated mice, treated with 10 mg/kg enzalutamide, was markedly suppressed (Day 30: 2.51-fold, *p* < 0.01), and this tumor growth suppression was even more pronounced in the mice inoculated with PC3_shGSE1 cells alone (Day 30: 4.67-fold, *p* < 0.001) or coupled with 10 mg/kg enzalutamide treatment (Day 30: 11.16-fold, *p* < 0.001) (Figure 7F). Concordant with our in silica and in vitro data, the post-study IHC staining of harvested tumor samples showed that, compared to the expression in the control samples, enzalutamide treatment, inoculation with PC3_shGSE1, or PC3_shGSE1 coupled with enzalutamide treatment, suppressed the GSE1, MDR1, and OCT3/4 protein expression levels, in increasing order of magnitude. However, TACSTD2 exhibited an opposite expression profile (Figure 7G). These data indicate that targeting GSE1 signaling suppresses metastatic and cancer stemness phenotypes and enhances sensitivity to abiraterone or enzalutamide in metastatic castration-resistant prostate cancer.

## 4. Discussion

Despite significant advances in diagnostic and therapeutic strategies in the last three decades, metastatic castration-resistant prostate cancer (mCRPC) remains a therapeutic enigma. In the last decade alone, the United States Food and Drugs Administration (US FDA) approved six new anticancer drugs for managing CRPC, with a seventh one ascribed a ‘breakthrough designation for accelerated development based on biomarker status’ [22], highlighting the clinical relevance and therapeutic indispensability of actionable biomarkers in the management of ‘difficult-to-treat’ and ‘quick-to-relapse’ malignancies, such as mCRPC. Premised on this understanding, we aimed to identify such objective indicators of cancerization, disease course, and therapy response in patients with advanced PCa, including mCRPC.

In the present study, we demonstrated (for the first time, to the best of our knowledge) that (i) an increased GSE1/TACSTD2 expression ratio marks patients with PCa and that (ii) the interaction between GSE1 and TACSTD2 drives metastatic disease, castration resistance, and disease progression in patients with PCa. We also provide preclinical evidence that (iii) inversely correlated GSE1 and TACSTD2 expression patterns determine the survival of patients with PCa and that (iv) the GSE1 and TACSTD2 signal interplay affects the clinical and immune statuses and (v) determines the therapy responses and clinical outcomes in patients with PCa. Of therapeutic relevance, we also demonstrate that (vi) the GSE1 and TACSTD2 interaction, or expression profiles, reflect abiraterone/enzalutamide drug resistance, androgen sensitivity, and castration resistance in patients with prostate cancer and posit that (vi) targeting GSE1 signaling suppresses metastatic and cancer stemness phenotypes and enhances sensitivity to abiraterone or enzalutamide in metastatic castration-resistant PCa in vitro and in vivo.

The observation that GSE1 is upregulated, while TACSTD2 is downregulated, in PCa, with an increased GSE1/TACSTD2 expression ratio defining patients with PCa (Figure 1), is not logically decoupled from the current dysregulated oncogene–tumor suppressor homeostasis paradigm associated with tumor initiation and progression. Our finding is, in part, corroborated by a recent report that the oncoprotein GSE1 is aberrantly expressed in breast cancer and implicated in the proliferation, migration, and invasion of breast cancer cells [8]. Similar to TACSTD2, the same study also suggested that hsa-miR-489-5p, an onco-miR, is a direct target of GSE1 and exhibits suppressed expression in breast cancer cells [8].

While this study attributes a tumor suppressor role to TACSTD2 in PCa, we do acknowledge that this finding contradicts the findings of Hsu EC et al., who conversely stated that TACSTD2 is overexpressed in CRPC, drives cancer growth, and induces the neuroendocrine phenotype [23]. We cannot fully explain this contradiction; however, we cautiously attribute this to the tumor heterogeneity, mutational status, and/or therapeutic context. We find it intriguing that it was posited that the oncogenic activity of TACTSD2 was mediated by the upregulation of PARP1 [23], considering that the overactivation of the full-length PARP1 induces rapid cellular energy depletion, eliciting a shift in the cell death continuum from apoptosis to necrosis or necroptosis in the presence of enhanced DNA damage [24]. The ensuing conundrum is that, even in the presence of mild DNA damage, cleaved PARP1 quells necrosis/necroptosis, promotes apoptosis, and prevents cell survival [24]. Thus, in the context of [23,24], the induction or overexpression of TACSTD2 would elicit cell death, consistent with the position of the current study. Aligned with the therapeutic context hypothesis, although PARP1 participates in the DNA repair process, excessive chemotherapeutic- and/or radiation-induced DNA damage prompts PARP1 (and, by inference, TACSTD2) overactivation, adenosine triphosphate (ATP) depletion, and cell death, secondary to bio-energetic collapse [25,26]. It is also probable that the tumor suppressor role of TACSTD2, documented in our study, is associated with the resulting suppression of the synthesis of intranucleolar ribosomal RNA and the nuclear translocalization of TACSTD2 in the presence of GSE1 downregulation [27].

Our present findings provide some rationale for looking outside the contemporary classification of TACSTD2 as an oncogene by highlighting several tumor suppressor traits that make TACSTD2 a putative, actionable biomarker that should be carefully considered when developing any meaningful anti-mCRPC therapeutic strategy. In agreement with Shen et al.’s propositions on ‘genes with both oncogenic and tumor suppressor functions’ [28], it is probable that non-silent, function-altering mutations, such as frameshift or point mutations in TACSTD2, are the principal drivers of the prevalent tumor-limiting and therapy-sensitizing TACSTD2 signaling in CRPC, documented herein. Interestingly, similar to that of TP53, the mode of TACSTD2 activation is quite unique, compared to that of most other tumor suppressors; 82.5% of TACSTD2 mutations are truncating/nonsense mutations, as a C → T at nucleotide 352 replaces the glutamine at codon 118 with a Q118X stop codon [29]. This facilitates the synthesis of a stable mutant TACSTD2 protein, which accumulates in the plasma membrane and nucleus, subsequent to the inhibition of GSE1 signaling in aggressive mCRPC cells exposed to antiandrogen therapy and/or ADT. This high frequency of the premature termination/truncation of translation is strongly analogous to “four well-established tumor suppressors (PTEN, TP53, FBXW7, and CDKN2A), and PPP2R1A, a central component of the protein phosphatase 2A (PP2A) complex that also functions as a tumor suppressor” [30], regardless of the difference in the mutation spectrum. Moreover, while several studies have reported the activation of protein kinase B (Akt) by TACSTD2, mostly in cancer cell lines, several others have also described the downregulation of Akt and MAPK/ERK pathways by TACSTD2 (reviewed in [11]), thus indicating that the prevalent effect of TACSTD2 may be context-dependent. This rationalization suggests that tumor heterogeneity, the therapeutic context, and the mutational status are therapeutically relevant, as they transcend the initial biological function attributed to TACSTD2 and can inform the discovery or development of new, highly efficacious anti-mCRPC therapeutic strategies.

Furthermore, concordant with our finding that the inversely correlated expression of GSE1 and TACSTD2 drives metastatic disease, castration resistance, and disease progression, as well as aids in determining the survival of patients with PCa (Figure 2 and Figure 3), some recent studies have indicated that high GSE1 expression was strongly associated with an advanced clinical stage, a high histological grade, a depth of invasion > 5 mm, lymph node metastasis, and decreased sensitivity to trastuzumab, as well as positively correlated with worse survival rates in patients with gastric cancer [9,10].

Consistent with contemporary knowledge that molecular interactions induced or facilitated by proline-rich motifs characterize many aspects of the immune response, and that these proline-rich factors mediate cell–cell communication, signal transduction, and antigen recognition [31], we also demonstrated that the GSE1 and TACSTD2 signal interplay affects the clinical and immune status and is indicative of the therapy responses and clinical outcomes in patients with PCa (Figure 4 and Figure 5). It is clinically relevant that *GSE1* expression was higher in iClusters 1, 3, and 5 than in iClusters 2 and 6, while conversely, *TACSTD2* expression was lower in iClusters 1 and 5 than in iClusters 2, 3, 4, and 6. Corroborating this finding, a recent report indicates that, in terms of the time to the BCR endpoint, primary tumors in ‘minimally altered cluster 2 had an extremely favorable prognosis’, in contrast to the ‘extremely unfavorable prognosis for the highly altered cluster 5 tumors’ [15]. In addition, our finding is consistent with outcome reports from the Cambridge cohort, wherein iCluster 2 and iCluster 1 clearly distinguished patient groups with better and worse prognoses, respectively, based on BCR survival data collected over 60 months [16]. In fact, the same study reported that ‘iClusters1 and 3 identified men with the highest risk of relapse more effectively than either elevated Gleason score (≥4 + 3), high PSA, extracapsular extension (ECE) or positive surgical margin (PSM), especially as about 6-in-10 of the iCluster1, and 8-in-10 of the iCluster3 patients progressed to recurrent disease’ [16]. It is, thus, conceivable that an elevated GSE1/TACSTD2 ratio better stratifies patients into good and poor outcome groups than PSA, ECE, and PSM. This study is currently ongoing.

Cancer is widely considered to be a systemic disease that induces a myriad of functional and constitutive changes to the host immune system as a whole. Based on the findings of the present study, we posit that GSE1–TACSTD2 signaling modulates existing host anticancer immunity through the reprogramming of the stromal and intratumoral infiltrating immune cell pool, as we demonstrated, to elicit an immune-excluded/suppressive TME, and subsequent treatment failure (for the GSE1^high^TACSTD2^low^ genotype), or immune-activated/reinvigorated hot tumors, resulting in durable remission (for the GSE1^low^TACSTD2^high^ genotype) [32].

Consistent with their demonstrated role in immunoediting (for the first time, to the best of our knowledge), we also provided some preclinical evidence that the GSE1/TACSTD2 ratio reflects abiraterone/enzalutamide and castration resistance in patients with PCa and that targeting GSE1 signaling suppresses metastatic and cancer stemness phenotypes and enhances sensitivity to abiraterone or enzalutamide in metastatic castration-resistant PCa in vitro and in vivo (Figure 6 and Figure 7). This is of translational relevance, especially considering that, despite the touted benefits of enzalutamide and abiraterone, only a subset of patients with CRPC respond to these treatments, with a rather disappointing increase in median PFS of < 6 months, relative to that with the standard of care and an almost universal acquisition of abiraterone or enzalutamide resistance [33]. In line with the demonstrated *GSE1^low^TACSTD2^high^* expression in enzalutamide-sensitive cases, *GSE1^high^TACSTD2^low^* expression in enzalutamide-resistant cases, and high GSE1/TACSTD2 ratio in resistant cases, we posit that, while the known mechanisms underlying abiraterone or enzalutamide resistance remain inconclusive and continue to evolve, it is probable that by binding to and suppressing the catalytic activity of TACSTD2, GSE1 upregulates intratumoral and/or systemic androgen biosynthesis and deregulates pathways that crosstalk with androgen receptor (AR) signaling, with a consequent amplification of cancer stemness and AR signaling, while concomitantly facilitating an immunosuppressive TME.

As is characteristic of many studies of this nature, the present preclinical study was limited by the need for the inclusion of more clinicopathological features of PCa, such as the Gleason grade, AR score, volume of metastatic disease, microsatellite instability (MSI) score, tumor mutation burden (TMB), and RAF1/BRAF status. As such, the data presented herein should be interpreted ‘as is’ and with cautious optimism.

## 5. Conclusions

Taken together, these data provide preclinical evidence of the oncogenic role of dysregulated GSE1–TACSTD2 signaling and show that the molecular, or pharmacological, targeting of GSE1 is a workable therapeutic strategy for inhibiting androgen-driven oncogenic signals, re-sensitizing CRPC to treatment, and repressing the metastatic/recurrent phenotypes of patients with PCa.

## Figures and Tables

**Figure 1 cancers-13-03959-f001:**
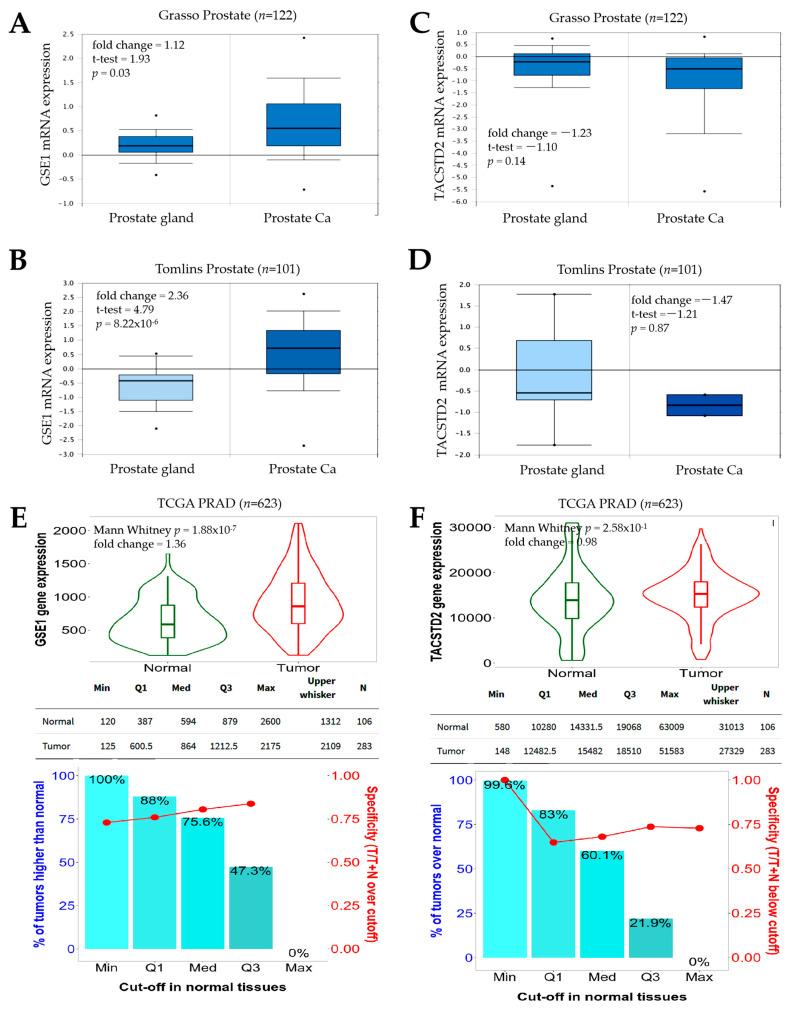
Increased GSE1/TACSTD2 expression ratio defines patients with prostate cancer. Box-and-whisker plots showing the differential expression of *GSE1* mRNA in prostate glands and prostate cancer in the (**A**) Grosso and (**B**) Tomlins Prostate cohorts. Box-and-whisker plots showing the differential expression of *TACSTD2* mRNA in prostate glands and prostate cancer in the (**C**) Grosso and (**D**) Tomlins Prostate cohorts. Violin plots showing the differential expression (**top**) and expression cut-off chart (**bottom**) of (**E**) *GSE1* and (**F**) *TACSTD2* in normal and tumor samples in the TCGA PRAD cohort. TCGA, The Cancer Genome Atlas; PRAD, Prostate Adenocarcinoma.

**Figure 2 cancers-13-03959-f002:**
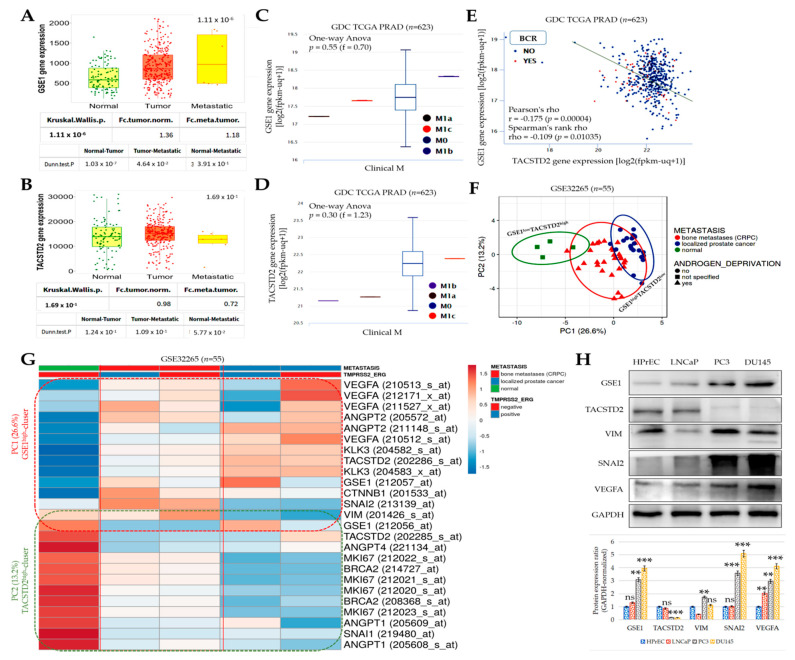
The interaction between GSE1 and TACSTD2 drives metastatic disease, castration resistance, and disease progression in patients with prostate cancer. Box-and-dot plots showing the differential expression of (**A**) *GSE1* and (**B**) *TACSTD2* genes in normal, tumor, and metastatic samples from the TCGA PRAD cohort. Box-and-whisker plots showing the correlation between (**C**) *GSE1* or (**D**) *TACSTD2* gene expression and clinical M stage in the TCGA PRAD dataset. (**E**) Graphical representation of the correlation between *GSE1* and *TACSTD2* gene expression levels. (**F**) Principal component analysis of the expression data for primary localized PCa vs. castration-resistant bone metastatic prostate cancer in the GSE32265 cohort. Unit-variance scaling was applied to rows; SVD with imputation was used to calculate principal components. X- and Y-axes show Principal Components 1 and 2, which explain 26.6 and 13.2% of the total variance, respectively. Prediction ellipses were computed such that, with a probability of 0.95, a new observation from the same group would fall inside the ellipse. N = 55 data points. (**G**) Expression heatmap showing the relationship between metastasis, the TMPRSS2-ERG fusion gene, and expression of GSE1, TACSTD2, KLK3/PSA, and biomarkers of angiogenesis, metastasis, and proliferation in the GSE32265 cohort. Columns with similar annotations were collapsed by taking the mean in each group. Rows are centered; unit-variance scaling was applied to rows. Both rows and columns were clustered using correlation distance and average linkage, with 25 rows and 5 columns. (**H**) Representative Western blot images of the differential expression of GSE1, TACSTD2, VIM, SNAI2, and VEGFA proteins in HPrEC, LNCaP, PC3, or DU145 cells. GAPDH served as a loading control. FC, fold change; meta, metastatic; norm, normal; M, distant metastasis; M0, no distant metastasis; M1a, distant metastasis to non-regional lymph node(s); M1b, distant metastasis to bone(s); M1c, distant metastasis to other site(s) with or without bone disease; BCR, biochemical recurrence; SVD, singular value decomposition. ** *p* < 0.01; *** *p* < 0.001; ns, not significant.

**Figure 3 cancers-13-03959-f003:**
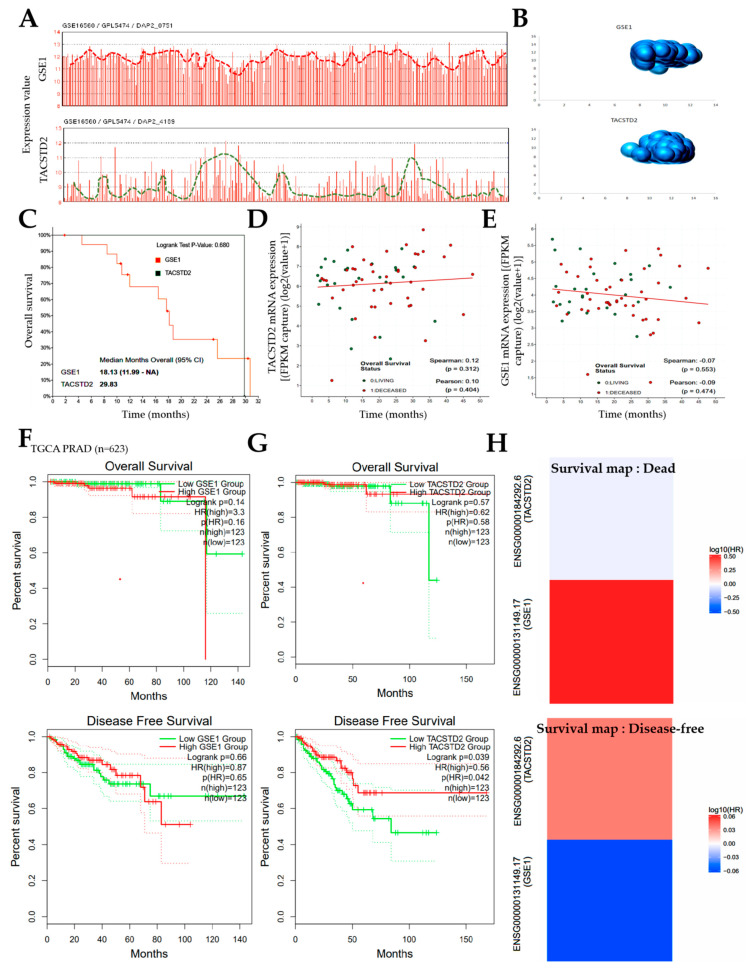
Inversely correlated GSE1 and TACSTD2 expression patterns determine survival of patients with prostate cancer. (**A**) Correlation histograms for *GSE1* and *TACSTD2* expression pattern recognition in the GSE16560/GPL5474 PCa disease progression dataset. (**B**) Bubble charts of the convergent expression profile of GSE1 and TACSTD2 in the GSE16560/GPL5474 PCa cohort. (**C**) Kaplan–Meier curve showing the differential effects of GSE1 and TACSTD2 expression on overall survival in the pooled prostate cancer (DKFZ, Cancer Cell (2018))/prostate adenocarcinoma (MSKCC/DFCI, Nature Genetics (2018))/metastatic castration-sensitive prostate cancer (MSK, Clin Cancer Res (2020))/metastatic prostate adenocarcinoma (SU2C/PCF Dream Team, PNAS (2019)) cohort. Scatter plots showing the correlation between (**D**) *TACSTD2* or (**E**) *GSE1* mRNA expression and survival time in the pooled cohort. Kaplan–Meier curve showing the effect of the differential expression of (**F**) GSE1 or (**G**) TACSTD2 on overall survival (**top**) and disease-free survival (**bottom**). (**H**) Survival maps showing association between *GSE1* or *TACSTD2* with death or disease-free survival.

**Figure 4 cancers-13-03959-f004:**
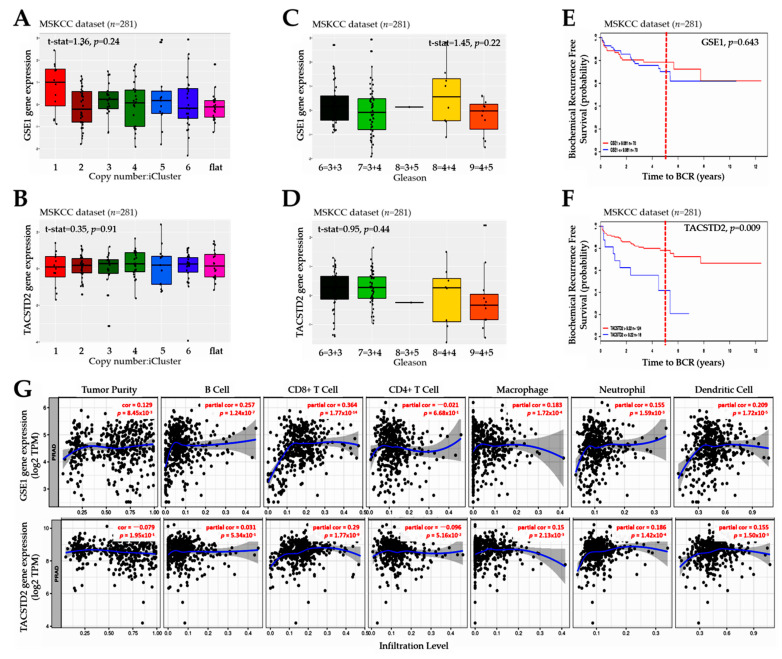
The GSE1 and TACSTD2 signal interplay affects the clinical and immune statuses of patients with prostate cancer. Box-and-whisker plots showing the relationship between (**A**) *GSE1* or (**B**) *TACSTD2* expression and copy-number-based iClusters in the MSKCC dataset. Box-and-whisker plots of the relationship between (**C**) *GSE1* or (**D**) *TACSTD2* expression and Gleason score in the MSKCC dataset. Kaplan–Meier curves showing the effect of the differential expression of (**E**) GSE1 or (**F**) TACSTD2 on biochemical recurrence-free survival. (**G**) Scatter-plot visualization of the tumor purity-adjusted correlation between GSE1 or TACSTD2 expression and immune cell infiltration levels in PRAD. The abundance of six immune infiltrates, namely, B cells, CD4+ T cells, CD8+ T cells, neutrophils, macrophages, and dendritic cells, were estimated using the TIMER algorithm. The gene expression levels against tumor purity are displayed in the left-most panel. Genes highly expressed in the microenvironment are expected to have negative associations with tumor purity, while the opposite is expected for genes highly expressed in the tumor cells.

**Figure 5 cancers-13-03959-f005:**
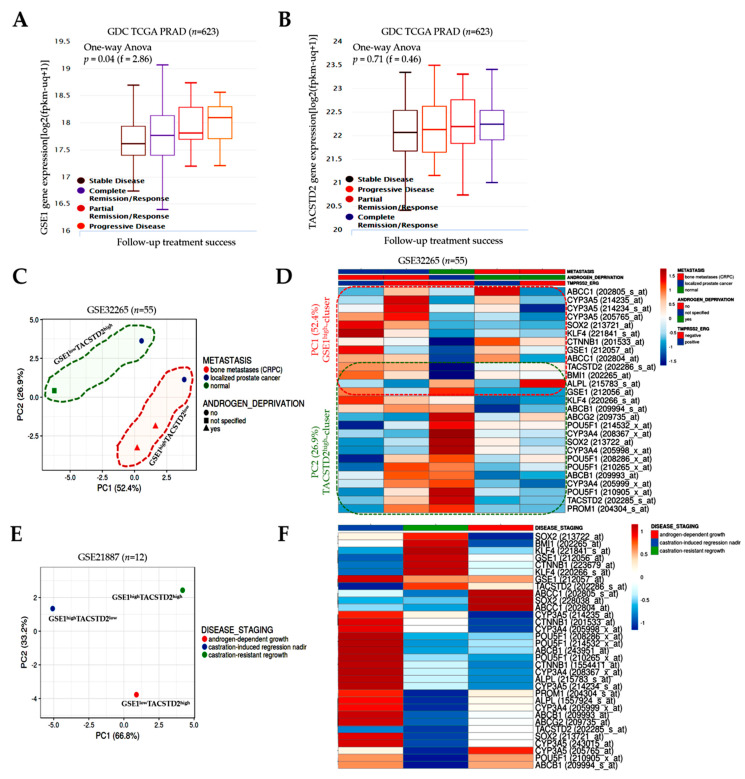
GSE1 and TACSTD2 expression profiles are indicative of therapy responses and clinical outcomes in patients with prostate cancer. Box-and-whisker plots showing how (**A**) *GSE1* or (**B**) *TACSTD2* expression affects follow-up treatment success in the TCGA PRAD cohort. (**C**) Principal component analysis of the expression data for primary localized PCa vs. castration-resistant bone metastatic prostate in the GSE32265 cohort. Columns with similar annotations were collapsed by taking the mean in each group. Unit-variance scaling was applied to rows; SVD with imputation was used to calculate principal components. X- and Y-axes show Principal Components 1 and 2, which explain 52.4 and 26.9% of the total variance, respectively. N = 5 data points. (**D**) Expression heatmap showing the association between metastasis, androgen deprivation, TMPRSS2-ERG fusion gene, and expression of GSE1, TACSTD2, biomarkers of drug metabolism/resistance, and cancer stemness in the GSE32265 cohort. (**E**) Principal component analysis of the expression data for the treatment of castration-resistant prostate cancer in the GSE21887 cohort. X- and Y-axes show Principal Components 1 and 2, which explain 66.8 and 33.2% of the total variance, respectively. (**F**) Expression heatmap showing the association between disease stage and expression of GSE1, TACSTD2, biomarkers of drug metabolism/resistance, and cancer stemness in the GSE21887 cohort.

**Figure 6 cancers-13-03959-f006:**
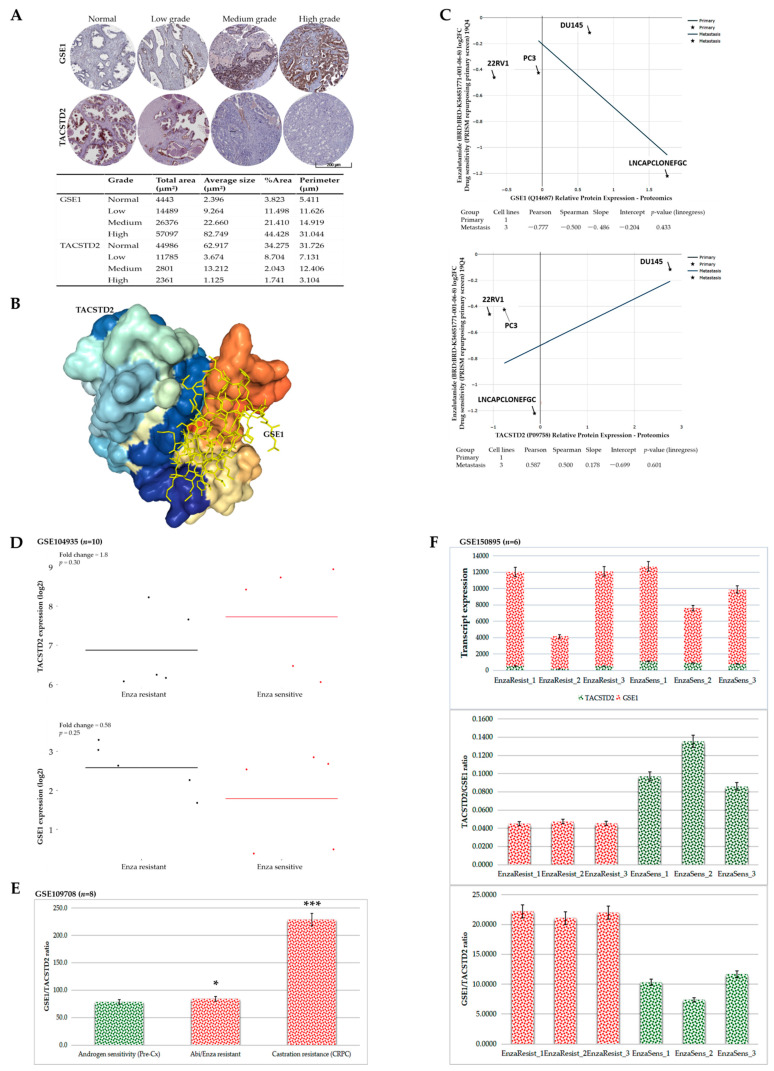
GSE1 and TACSTD2 interaction or expression profiles reflect abiraterone/enzalutamide drug resistance, androgen sensitivity, and castration resistance in patients with prostate cancer. (**A**) Representative IHC staining photomicrographs (**top**) and quantitative chart (**bottom**) of the differential expression of GSE1 or TACSTD2 protein in normal, low-grade, medium-grade, and high-grade PCa samples from the TMU-SHH cohort. (**B**) Molecular docking showing the direct interaction between TACSTD2 and GSE1. (**C**) Graphical representations of the association between GSE1 (**top**) or TACSTD2 (**bottom**) protein expression and enzalutamide sensitivity in 22RV1, DU145, PC3, and LNCaP clone FGC cell lines. (**D**) Line and dot plots of the differential expression of TACSTD2 (**top**) and GSE1 (**bottom**) in enzalutamide-resistant and enzalutamide-sensitive samples from the GSE104935 cohort. (**E**) Histograms showing the association between GSE1/TACSTD2 ratio and pre-castration androgen sensitivity, abiraterone/enzalutamide resistance, or castration resistance in the GSE109708 cohort. (**F**) Histograms showing *GSE1* and *TACSTD2* transcript expression (**top**), TACSTD2/GSE1 ratio (**middle**), and GSE1/TACSTD2 ratio (**bottom**) in enzalutamide-resistant or sensitive samples from the GSE150895 cohort. * *p* < 0.05; *** *p* < 0.001.

**Figure 7 cancers-13-03959-f007:**
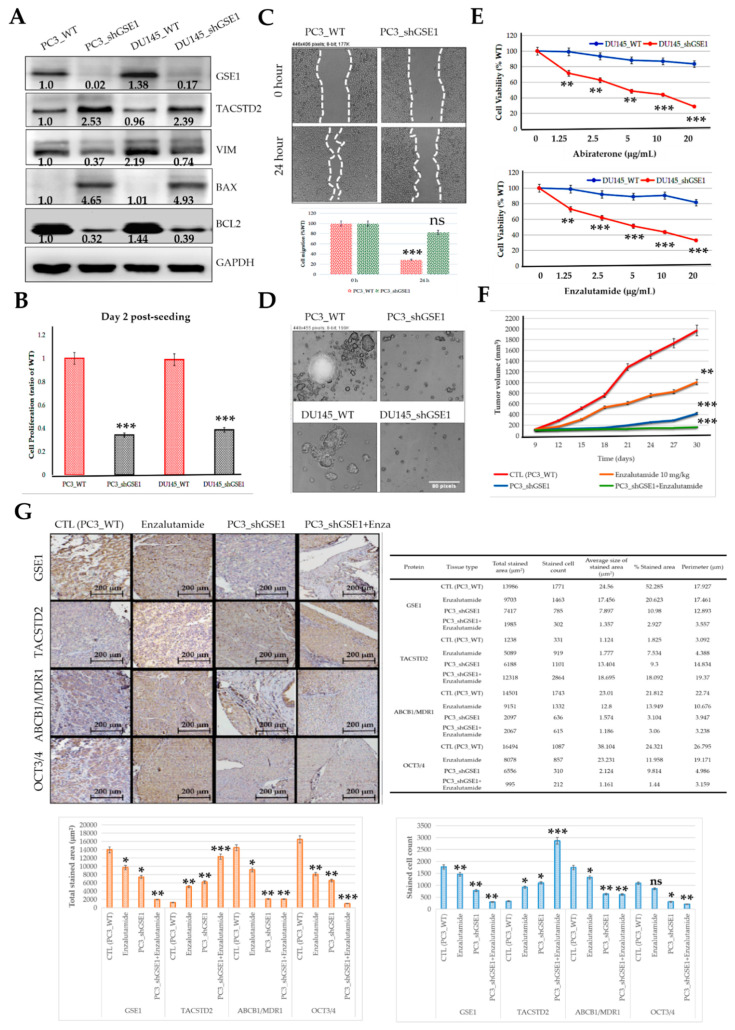
Targeting GSE1 signaling suppresses metastatic and cancer stemness phenotypes and enhances sensitivity to abiraterone or enzalutamide in metastatic castration-resistant prostate cancer in vitro and in vivo. (**A**) Representative Western blot images showing the differential expression of GSE1, TACSTD2, VIM, SNAI2, BAX, and BCL2 in PC3_WT, PC3_shGSE1, DU145_WT, and DU145_shGSE1 cells. GAPDH served as a loading control. (**B**) Histograms comparing PC3_WT, PC3_shGSE1, DU145_WT, and DU145_shGSE1 cell proliferation on Day 2 post-seeding. (**C**) Representative photomicrographs and histograms of migration by PC3_WT or PC3_shGSE1 over 24 h. (**D**) Representative images of the effect of shGSE1 on the formation of tumorspheres in PC3 or DU145 cell lines. (**E**) Line graphs showing the effect of 1.25–20 mg/mL Abiraterone (**upper**) or Enzalutamide (**lower**) on DU145_WT and DU145_shGSE1 cell viability. (**F**) Line graph showing the time-lapsed tumor volumes in mice inoculated with PC3_WT or PC3_shGSE1, with or without 10 mg/kg enzalutamide treatment. (**G**) Representative IHC staining photomicrographs (**upper left**), quantitative chart (**upper right**), total stained area histograms (**lower left**), and stained cell count histograms (**lower right**) of the differential expression of GSE1, TACSTD2, MDR1, and OCT3/4 proteins in tumor samples harvested from mice inoculated with PC3_WT or PC3_shGSE1, with or without 10 mg/kg enzalutamide treatment. CTL, control; WT, wild-type; * *p* < 0.05; ** *p* < 0.01; *** *p* < 0.001; ns, not significant.

## Data Availability

The datasets used and analyzed in the current study are publicly accessible, as indicated in the manuscript.

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
