# Peer review of "Genetic Suppressor Element 1 (GSE1) Promotes the Oncogenic and Recurrent Phenotypes of Castration-Resistant Prostate Cancer by Targeting Tumor-Associated Calcium Signal Transducer 2 (TACSTD2)"

_cancers, 2021, doi:10.3390/cancers13163959_

Round 1

Reviewer 1 Report

The manuscript 'Genetic Suppressor Element 1 (GSE1) promotes the oncogenic and recurrent phenotypes of Castration-Resistant Prostate Cancer by targeting Tumor Associated Calcium Signal Transducer 2 (TACSTD2)' by Bamodu et al. investigates the oncogenic role of dysregulated GSE1-TACSTD2 signaling in Castration-Resistant-Prostate Cancer and highlights a potential role of this signaling axis in pharmacological treatment.  

The manuscript is well written and provides valuable insights into PCa development. The authors used a large number of methods to study the GSE1 signaling axis including histological and biochemical assays as well as in vitro and in vivo experiments. 

I have only a minor comment regarding consistency of gene spelling. Human genes are written in italic and capital letters. Therefore, I would like to ask the authors to correct this as not all genes are written in this style.

Reviewer 2 Report

An article of Oluwaseun Adebayo Bamodu and co-workers relates to the GSE1 and TACSTD2 proteins and their role in prostate cancer development and progression. Though the work is of interest a number of important/crucial and a huge amount of minor issues are reported.

Please see comments in line per line order below: 

1) Line 37, "Aided by analyses" needs English rephrasing like "using large cohort.."

2) Line 41: what TACSTD2 stands for ? Why you uncover GSE1 but not TACSTD2 ?

3) Line 41: "More so" needs to be replaced by some more English phrase like "moreover", "further", "in addition", etc....

4) Line 46: "preempts" = "predicts" ? and everywhere throughout the text. 

6) Line 59:  Is everyone supposed to know what ICD: C61 means ?

7) Line 59-61: What country or worldwide this statistics concerns ?

8) Line 63: "triad" is not that pertinent in this sentence.

9) Line 63: "to pose significant challenge", seems "to create" is much better.

10) Lines 64-69 : this sentence is really hardly written and needs paraphrasing.

11) I'm sorry but it's not possible to read your text in English at the introduction section and extensive editing of English language and style is strongly required. Besides you largely use a discussion elements in your introduction so it's not introduction, but discussion session.

12) Line 145: "confluence ≥98%" is such your cells are confluent, which bias all your experiments if it's true.

13) Line 156: What " (h) " stands for ?

14) Line 155: For how long cells were transfected ? Neither in the Figure 7A nor anywhere is not possible to find this time and its justification. The word "Stably-transfected" is met only once in the text, while shRNA can be also used as a transient transfection.

15) Line 165: it's a weird way to present " 0.3 × 104" instead of  3 × 10E3...

16) Line 164: cell survival is not proliferation. What authors do are both cell survival methods (SRB and alamarBlue™).

17) Line 168: it's weird to say:  "normal: Gleason score (GS) ≤ 5". It's not normal by definition. You have to omit "normal".

18) Line 21': "Scratch-wound healing migration assay" = just "wound healing assay". 

19) Line 233: how authors injected cells in complete growth medium without adding a Matrigel to maintain tumor cell growth ?

20) Why BALB/c-nu mice were used instead of other nu/nu or scid mice conventional in the field ?

21) Line 235: randomly placed

22) Line 236: into the control..

23) What was a rationale of using Enzalutamide against castration-resistant PC3 cells ?

24) Line 239: "positive: PC-3_WT or negative: PC-3_shGSE1". There's no negative control here. Negative control means cell-free medium injection.

25) Line 242: phrase "humanely sacrificed" is extremely weird. Just sacrificed should be preffered.

26) Line 243: "carefully analyzed", what do you mean ?

27) Line 243: "and samples used" ... what samples ? samples of tumors ??

28) Line 251: "aided" to be changed by assisted ?

29) Line 257:  phrase "defines patients with Prostate Cancer." means nothing.

30) Line 260:  exhibited significantly higher expression of GSE1 mRNA (fold change = 1.12, p = 0.03). Fold 1.12 is at the error limit level. What was a cut-off level ?

31) Line 261: "reanalysis" What do you mean ?

32) line 269: "was significantly upregulated (1.36-fold, p = 1.88e-07), tacstd2
gene expression was downregulated (0.98-fold, p = 2.58e-01)". Is significantly means "statistically significant" ? I'm sorry, but 1.36, nor 0.98 fold changes are barely significant...

33) Line 270:  "normal samples". What are "unnormal samples" ?

34) Line 272: "defines patients with PCa", as I mentioned above...

35) Line 320: are PC3 cells PSA positive ?

36) Figure 2H: Protein sizes, protein ladder are missing. Not acceptable in this form.

37) Figure 2G and H the police size sometimes is too small to be readible.

38) Line 353: not appropriate verb using: "we probed the GSE16560/GPL5474 PCa disease progression dataset "

39) Line 402: "More so"..., line 459 as well.

40) Line 429: not appropriate word using:  "has been implicated in cancer aggression".

41) Line 430: "has also been touted as a..." not a scientific language.

42) Line 452: "we further probed this possibility". We do not probe a possibility.

43) Line 478: Would you be so kind explaining "regrowth" & "inference recurrence" please?

44) Line 478: Would you be so kind explaining "nadir group" please?

45) Line 509: "normal samples" doesn't exist in science. Control samples, samples from healthy subjects, etc...

46) Line 515: "binds with" not correct, "binds to" is correct.

47) In the Figure 6A, is only area of staining calculated ? What about intensity score ? How authors explain that huge difference in the intensity of the signal with the very low expression changes (1.12-1.36 and 0.98) ?

48) Line 560: No real GSE1 and TACSTD2 interaction was shown by authors, only in silico modeling. The entire phrase 560-563 is a pure speculation.

49) Results from Figure 7A suggest that no interaction between GSE1 and TACSTD2 is mostly probable since their expression are inversely correlated ! More than interaction, they seem to suppress each other !

50) Lines 566-568: authors show no proliferation in this study, only cell survival.

51) Figure 7C: would healing is not a direct migration test since the effects of cell proliferation are hidden inside. Other methods like Boyden chamber assays should be used to talk about migration.

52) Figure 7E: Nonsens! How authors explain the effect of the anti-androgen drugs in androgen/castration-independent cell lines !!!

53) Idem, Figure 7F: Nonsens! as in the comment below.

Reviewer 3 Report

In this study, the authors investigated the role of the GSE1 molecule in the prognosis, therapy resistance, metastasis and disease recurrence in prostate cancer patients. 

They showed that the GSE1-TACSTD2 ratio can define the patient prognosis and survival and that GSE1 is a potential biomarker to target in prostate cancer.

The results are robust and corroborate the conclusion of the work.

Some minor comments, to increase the quality of the work for publication, especially with regard to its introduction and a few points in writing are presented below. There is also an excess of phrases that lack references.

Minor comments:

The introduction is well written, but there is no link between GSE1 and TACSTD2. A paragraph explaining the relationship between the two molecules would be important to better explain the authors' reasoning and introduce the work.

Line 64 - please add the reference.

Line 88 - please add the reference.

Line 92 - please add a reference.

Line 97-100 - the authors mentioned the existence of some reports showing that the GSE1 is overexpressed in patients with breast cancers [...], but only one reference is presented. Other references should be added to reinforce the hypothesis of the work.

Line 114 - please add a reference.

Reviewer 4 Report

The authors examine the role of GSE1 in prostate cancer. Their methodology is appropriate for a pre-clinical study and their systematic approach to analysis is sound. They found an increase in GSE1/TACSTD2 ratio was predictive of prostate cancer compared to normal tissue and a higher expression of GSE1 in metastatic disease as compared with tumor and TACSTD2 lowest in metastatic disease. Although these results are interesting from a biological perspective, they are unlikely to have an impact on disease management for some time. However, I believe the study is well designed, well annotated and acceptable for publication. My comment to the authorship would be they do not include acknowledgement of the considerable limitations to their analysis, the foremost of which is it's retrospective design and their necessary lack of inclusion of clinical features of cancer such as gleason grade or volume of metastatic disease as covariates in their analyses. Otherwise, the effects of metastases directed therapy cannot be examined in their cohort. I would emphasize that the authorship should also be a little more cautious in word choice throughout the discussion. The data are interesting but do not prove anything. They should openly acknowledge this.

Round 2

Reviewer 2 Report

Dear Authors,

Thank you for your revision of my comments on your paper.

I do appreciate your work and apologize for my losing of patience because of the low quality and not comprehensible English style of your paper. I did spend several days trying to understand what you wished to say and what you showed in your figures. Unlike my colleagues, who have found your manuscript « acceptable » or « minor revision », I did my work with conscience, reading your paper from A to Z so at the end I felt myself more co-author than Reviewer. I really tried to help you to improve your manuscript, instead of refusing it immediately, but you did not appreciate it.

You have preferred to insult me and to review my comments instead of understanding what I have written and why I have provided such comments. I have not seen who you are nor the comments of my colleagues Reviewers. I wanted always to be objective, and I was.

Now you can complain to whatever and whomever you want, I can again undersign under each my comment I did for you to improve your manuscript. And for sure, after your misrespecting reply, I will not help you anymore. As a Reviewer of the journal « Cancers » I cannot accept your manuscript in its current form.

Thank you for your understanding.
